# Impact of Magnesium Supplementation in Muscle Damage of Professional Cyclists Competing in a Stage Race

**DOI:** 10.3390/nu11081927

**Published:** 2019-08-16

**Authors:** Alfredo Córdova, Juan Mielgo-Ayuso, Enrique Roche, Alberto Caballero-García, Diego Fernandez-Lázaro

**Affiliations:** 1Department of Biochemistry and Physiology, Faculty of Health of Sciences, University of Valladolid Campus de Soria, 42003 Soria, Spain; 2Department of Applied Biology-Nutrition and Institute of Bioengineering, Alicante Institute for Health and Biomedical Research (ISABIAL Foundation), 03550 Alicante, Spain; 3CIBERobn (Fisiopatología de la Obesidad y la Nutrición CB12/03/30038) Instituto de Salud Carlos III, 28029 Madrid, Spain; 4Department of Anatomy and Radiology, Faculty of Health of Sciences, University of Valladolid, 42004 Soria, Spain; 5Department of Cell Biology, Histology and Pharmacology, Faculty of Health of Sciences, University of Valladolid, 42004 Soria, Spain

**Keywords:** cycling, exercise, muscle damage, performance, supplementation

## Abstract

Magnesium is a cofactor of different enzymatic reactions involved in anabolic and catabolic processes that affect muscular performance during exercise. In addition, it has been suggested that magnesium could participate in maintaining muscle integrity during demanding effort. The main purpose of this study was to analyze the effects of magnesium supplementation in preventing muscle damage in professional cyclists taking part in a 21-day cycling stage race. Eighteen male professional cyclists (*n* = 18) from two teams were recruited to participate in the research. They were divided into 2 groups: the control group (*n* = 9) and the magnesium-supplemented group (*n* = 9). The supplementation consisted of an intake of 400 mg/day of magnesium during the 3 weeks of competition. Blood samples were collected according to World Anti-Doping Agency rules at three specific moments during competition: immediately before the race; mid competition; and before the last stage. Levels of serum and erythrocyte magnesium, lactate dehydrogenase, creatinine kinase, aspartate transaminase, alanine transaminase, myoglobin, aldolase, total proteins, cortisol and creatinine were determined. Serum and erythrocyte magnesium levels decreased during the race. Circulating tissue markers increased at the end of the race in both groups. However, myoglobin increase was mitigated in the supplemented group compared with the controls. We conclude that magnesium supplementation seems to exert a protective effect on muscle damage.

## 1. Introduction

In sports, an inadequate body mineral composition can lead to decreased performance. Magnesium (Mg) is an intracellular cofactor involved in more than 300 enzymatic reactions, in both anabolic and catabolic processes. For this reason, low intracellular levels can directly affect muscular performance, especially during exercise [1,2,3]. As a result, magnesium deficiency produces muscle weakness and damage which is reflected by an increase of circulating muscle damage markers, compromising subsequent recovery [4,5,6]. This could be a point of particular attention in extensive aerobic exercises in which completion is performed through consecutive stages with short recovery periods.

Since Mg is an intracellular mineral, it is difficult to assess if Mg circulating levels could reflect total tissue contents for this mineral, and at the same time, the effects of supplementation. The compartmental nature of Mg might be the explanation for the divisive results obtained in different studies. In this context, Stendig-Lindberg et al. [7] have suggested that increased circulating Mg levels after demanding efforts might reflect a release from damaged muscle. Therefore, no changes in circulating Mg status during exercise might be related to tissue and cell protection in athletes [8,9]. However, Mg can be profusely lost through transpiration [10], making it difficult to establish correct assessments regarding Mg distribution in the different body compartments during supplementation protocols. In addition to this instrumental limitation, other factors that can complicate the interpretation of results include a variety of experimental designs, different exercise intensities and duration, timing for obtaining blood samples, environmental conditions and stress during performance [2,3,5,6,11,12,13]. Correct Mg assessment in the body requires specific overloading protocols that are tedious to perform during competition [3,13]. Alternatively, Mg can be determined in circulating cells as a reflection of how compartmental Mg varies during exercise and the likely effect of supplementation. However, the Mg content in the different tissues varies, making it difficult to interpret results.

Recently, we have studied the effects of Mg supplementation on muscle damage markers in basketball players over a full season, concluding that Mg supplementation could contribute to preventing muscle damage [14]. Basketball is characterized by the combination of eccentric and concentric muscular contractions with sequences of intense activities (sprinting, shuffling, jumping) based on the anaerobic-capacity as a determinant of performance in high-level players [15]. However, in endurance sports such as cycling, which has different determinants (continuous concentric muscular contraction under extensive aerobic conditions), Mg supplementation has not been extensively tested [16]. According to the authors′ knowledge, only a few studies have reported the behavior of Mg in athletes during a long-term endurance exercise.

In view of this information, the aim of the present study was to examine the effects of Mg supplementation in preventing muscle damage in professional cyclists taking part in a 21-day cycling stage race (“Vuelta a España”). This was assessed by determining circulating Mg and Mg content in erythrocytes in order to reflect changes in Mg located in compartments. Muscle integrity and adaptation to demanding efforts were monitored by determining specific circulating markers.

## 2. Materials and Methods

### 2.1. Participants

Eighteen professional male cyclists from two different professional teams participated in this study. The athletes competed in a 3-week cycling stage race known as “Vuelta a España” that covers around 3300 km. Anthropometric and functional characteristics of the participants before the race are shown in Table 1 and Table 2, respectively. The examination made by the “Union Cycliste Internationale” (UCI or International Cyclist Union) before starting the competition confirmed the absence of pathologies among participants. None of the cyclists were taking banned drugs or medications, nor did any cyclists test positive in the routine doping tests performed before and during the race according to the World Anti-Doping Agency (WADA).

Taking into account that the participants had similar objectives and participated in the same race, all followed a very similar dietetic and training program to prepare for the race under the control of the team physician in collaboration with one of the authors of the study, who was a clinician. The participants were informed of the experimental procedures, risks, and benefits, and voluntarily signed a written consent. The study was designed according to the Declaration of Helsinki for experiments with human beings and was approved by the local Ethics Committee of the University of Valladolid.

### 2.2. Experimental Protocol and Assessment Plan

A randomized study design comparing control (no supplement) to Mg supplementation was applied to analyze the effects of oral magnesium supplementation on serum magnesium (Mg), erythrocytic magnesium (e-Mg), haematological parameters and inflammation/muscle damage biomarkers.

Participants were assigned to one of two groups: (a) the control group (CG, *n* = 9) that did not receive oral magnesium supplementation; or (b) the magnesium-supplemented group (MG, *n* = 9) in which they received oral magnesium supplementation (400 mg/day). An independent statistician generated the random allocation sequence. The cyclists were tested in three specific points during the study: at baseline one day before the start of the race (T1); before the start of the 10th stage (T2); and before the last stage (20th) of the race (T3). The MG received an oral supplementation of 400 mg/day of pure Mg (magnesium 400 supra Kapsel^®^, Sanct Bernhard, Barcelona, Spain), every morning at breakfast over the course of the whole race. The bioavailability of Mg from this presentation was estimated as 35%, according to the supplier information [17]. During the competition, the research team ensured constant circadian rhythms in terms of nutrition, hydration, timing of food intake and sleep were maintained, which is a normal practice in cycling competitions.

### 2.3. Dietary Assessment

The dietitian of the team recorded the daily food and fluid intake of the cyclists during the 21-day study. The EasyDiet^©^ software, available online (https://www.easydiet.es/), was used to calculate nutrient composition and energy intake from food and drinks consumed by the cyclists. This software package was developed by the Spanish Centre for Higher Studies in Nutrition and Dietetics (CESNID), which is based on Spanish tables of food composition [18]. The mean Mg consumption in the diet of participants in our study was 247 ± 5.3 mg/1000 kcal, for an average energy intake estimated of 5000 kcal/day (75% carbohydrates, 14% proteins and 11% fat) [19]. Also, all participants (*n* = 18) received 10 mg of folic acid/day (Interpharma, Barcelona, Spain), 1 g of vitamin C/day (Bayer Redoxon^®^, Barcelona, Spain), 1000 µg of vitamin B12/day (Solgar S.L., Madrid, Spain), branched chain amino acids (3600 mg of l-leucine/day, 900 mg of L-isoleucine/day and 900 mg of l-valine/day, Quamtrax^®^, Madrid, Spain) and 1 g of glutamine/day (Gold Nutrition, Manique, Portugal).

### 2.4. Anthropometry

Anthropometric data (Table 1) were taken by an ISAK (International Society for Advancement in Kinanthropometry) level 3 anthropometrist, following the standard procedures at the beginning of T1. The height and body mass technical error of measurement (TEM) was less than 0.02%. For skinfolds, TEM was less than 2.6% in all cases. The height (cm) was measured with a SECA^®^ measuring rod (Barcelona, Spain), with a precision of 1 mm (range: 130–210 cm). Body weight (kg) was assessed using a SECA^®^ scale, with a precision of 0.1 kg (range: 2–130 kg). The 6 skinfolds (triceps, abdominal, supraspinale, subscapular, front thigh and medial calf) were measured with Harpenden skinfold calliper (CMS instruments, London, UK), with a precision of 0.2 mm.

### 2.5. Blood Collection and Analysis

Blood extraction and transportation were performed according the UCI and WADA guidelines (www.ama-wada.org). All samples were collected in basal conditions after a 10–12 h overnight fast at T1, T2 and T3. Blood samples (around 15 mL) were obtained at 8:30 a.m. from the antecubital vein with the subject seated in a comfortable position using Vacutainer tubes. Blood was distributed in one tube with gel and clot activator (10 mL) to obtain serum and another ethylenediaminetetraacetic acid (EDTA) tube (3–5 mL) to obtain plasma.

Immediately after filling, EDTA tubes containing blood were inverted 10 times and stored in a sealed box at 4 °C. Controlled temperature was assured during transportation: the specific tag (Libero Ti1, Elpro, Buchs, Switzerland) was used for temperature measurement and recording. The EDTA anticoagulated blood was processed according to UCI and WADA recommendations [20]. Plasma was obtained by centrifugation at 2000 rpm for 15 min. The outermost layer (plasma) was extracted using a Pasteur pipette, transferred to a sterile tube and stored at −20 °C until analysis.

Serum Mg was determined using a Perkin Elmer 272 by flame atomic absorption spectrometry (FAAS) in flame emission mode. The e-Mg was determined in hemolyzed blood obtained by dilution in deionized water, mixed with a vortex and frozen. After Mg determination in whole blood and plasma, e-Mg concentration was calculated as: (whole blood Mg-plasma Mg) × (1 − Haematocrit)/Haematocrit. Red blood cell (RBC), haemoglobin (Hb), and haematocrit (Hct) levels were determined on a Coulter Counter (Sysmex XE-2100). Total serum proteins, creatinine, creatin-kinase (CK), lactate dehydrogenase (LDH), aspartate transaminase (AST), alanine transaminase (ALT) and aldolase (ALD) were determined by standard methods using an autoanalyzer Hitachi 917 (Tokyo, Japan). Myoglobin (Mb) assessment was performed using a chemiluminescence immunoassay.

### 2.6. Performance Test: Maximum Oxygen Volume and Peak Power

At the beginning (T1) and at the end of the study (T3), maximum oxygen volume (VO_2_max) and peak power (Pmax) were determined in all participants by using an incremental test to exhaustion on a calibrated cycle-ergometer (Lode Excalibur Sport Cycle-Ergometer, Groningen, The Netherlands). Relationship Pmax/weight (IEmax) was calculated as Watt/kg. Participants began cycling at 100 Watt for 5 min with increases in intensity of 50 Watt every 2.5 min until exhaustion. During the VO_2_max testing, expired air was collected and analyzed using a calibrated metabolic cart (Vmax 29, SensorMedics, Yorba Linda, CA, USA).

### 2.7. Session Rating of Perceived Exertion

The physical effort intensity was assessed by session rating of perceived exertion (RPE) using a CR-10 scale, as previously described by Borg and adapted by Foster [21]. This method was used as an indicator of subjective perception of exercise intensity for monitoring competition load. Cyclists answered a simple question: “How was your workout (race)?” Answers were collected 30 min after cyclists crossed the finish line on the first and the last days of the race.

### 2.8. Statistical Data Analyses

Statistical analyses were performed using the IBM Statistical Package (SPSS Version 24). Data were expressed as mean ± standard error of the mean ($\bar{x}$ ± SEM). After checking the normal distribution using the Shapiro-Wilk test, the independent student′s t test was used for assessing differences between groups (CG vs. MG) at each time point (T1, T2 or T3) on hematological parameters, circulating enzymes and cortisol levels. Additionally, a two-way repeated measure analysis of variance (ANOVA) was carried out by Greenhouse-Geisser test to check the existence of an interaction effect (time × group) between CG and MG groups on hematological parameters, circulating enzymes and cortisol levels along the study (from T1 to T3). Likewise, a one-way repeated measure ANOVA was carried out on biochemical parameters (minerals, circulating enzymes, muscle damage markers and stress markers) by Greenhouse-Geisser test, during the different phases of the study, independently in each group. To determinate differences among periods of study, post hoc Bonferroni′s test was applied. Finally, bivariate correlations between Δ(T1 respect toT3) e-Mg during study and Δ(T1 respect to T3) circulating enzymes were tested using the Pearson rank order correlation test after calculating Δ(T1 respect to T3) = ((T3 − T1)/T1) × 100. A value of *p* < 0.05 was considered as significant.

## 3. Results

Table 1 and Table 2 depict anthropometric and functional characteristics of participants at T1 and T3. Although no significant differences between groups were found, Pmax, IEmax and VO_2_max were slightly higher in the MG compared to the CG at the end of the study (T3 in Table 2).

Figure 1 shows results regarding the evolution of serum Mg and e-Mg levels throughout the study (T1, T2 and T3) in both the CG and MG groups. At the beginning (T1), both groups displayed similar serum Mg and e-Mg levels, with both parameters decreasing significantly throughout T2 and T3. However, the decrease was significantly more pronounced in the CG when compared to the MG.

Table 3 displays hematological parameters during the 3 moments of study in both groups (MG and CG). White blood cells showed significantly higher values in both groups at the end of study compared to the baseline point (T1). Platelet number decreased significantly at the halfway point of the race (T2 vs. T1) in both groups, but recovered to initial values at the end of the race. Red blood cells did not present significant changes over the study in both groups, although a significant decrease in hemoglobin was observed at the halfway point of the race, but by the end of the race this had recovered to near the values observed at T1. Hematocrit followed the tendency observed for red blood cells, with no significant changes over the study. The decrease in hemoglobin could be related to a tendency to develop iron deficiency when performing demanding efforts, such as has been observed in this cycling race. The possibility of iron supplementation to avoid the decrease in hemoglobin has been described in a previous published study by our group [22].

Table 4 provides the levels of circulating markers and hormonal parameters at the 3 points (T1, T2 and T3) of study in both groups (MG and CG). In general, enzyme markers and cortisol did not present significant changes throughout the study, although they followed a tendency to increase at the end of the race. This increase was not significant when comparing both groups (CG vs. MG). However, other muscle markers, such as creatinine and Mb, increased significantly in both groups at the end of the race (T3). In addition, Mb displayed significant higher values in the CG compared to the MG at the halfway point and at the end of the race. Finally, total circulating proteins presented a significant decrease at the halfway point and at the end, compared to T1.

Of all protein markers, CK and Mb display the shortest half-lives in circulation (around 24 h). Therefore, these proteins are strong indicators of acute muscle damage that frequently occur during extremely demanding efforts such as stage cycling races, such as the “Vuelta a España”. Since circulating Mb decreased significantly in the MG compared to the CG, we wanted to verify whether there was a correlation between this marker and e-Mg. Even though e-Mg is not exactly an indicator of intramuscular Mg, it could reflect the release of this ion from intracellular compartments since muscular biopsies cannot be performed during the race. Figure 2 shows a correlation between e-Mg and Mb changes during the study (Figure 2A). A similar correlation was noticed between e-Mg and CK changes (Figure 2B), although CK did not present significant differences when comparing MG to CG (Table 4).

Finally, the mean scores obtained on session-RPE were 7.1 ± 2.0 and 7.2 ± 1.9 on the first day, and 8.1 ± 1.7 and 8.2 ± 1.9 on the last day of competition in CG and MG respectively, with no significant differences between the two groups during the study.

## 4. Discussion

The recommended dietary allowance (RDA) of Mg could be insufficient for athletes [23], since athletes are especially vulnerable to inadequate magnesium [24]. Cyclists are one of these susceptible populations because they undergo a significant decrease in body compartments that store Mg. In this context, body fat percentage, fat mass and upper arm fat mass significantly decreased after a long cycling stage race [25]. For this reason, we decided to test the effects of Mg supplementation on performance and muscle integrity in cyclists. Recently, we reported that oral supplementation with 400 of lactate Mg prevented tissue damage in basketball players [14].

Unlike basketball players, the population in the present study mainly performed aerobic effort with anaerobic peaks in particular moments of the race, such as sprinting and climbing. In addition, the changes in e-Mg allowed us to confirm the release from body compartments. A muscle biopsy together with an overloading protocol could be a more realistic approach to analyze Mg redistribution between body compartments, but these invasive protocols cannot be performed during such a demanding race as “Vuelta a España”. Therefore, serum Mg determination is not sufficient to paint a full picture of redistribution of Mg from one storage area (release) to an active site (muscle or adipose tissue), as previously documented [1,11,23,26,27,28]. Also, we have to consider the losses through sweat and urine, which are estimated at 10–20% [3].

An interesting finding was a negative correlation between e-Mg values with the most important muscle damage biomarkers (Mb and CK) (Figure 2). In this case, the cyclists with a lower release of Mg from the erythrocyte compartment (supplemented group) presented a better integrity of the muscle compartment. This suggests that the dynamics of e-Mg releases during long demanding efforts could be indicative of muscle status. In addition, this finding might suggest the importance of Mg on muscle function. In this sense, different studies in animal models have documented that disturbances on blood and tissue Mg levels play a role in a variety of muscle disorders mediated by oxidative stress, including myopathy, ultrastructural changes and skeletal muscle injuries [23,29,30,31]. In addition, in humans, systemic inflammation, distorted neuromuscular function and muscle damage have been inversely associated with low dietetic Mg intakes and directly with low Mg body levels [23,32]. In this context, increased extracellular Mg concentrations inhibit the generation of free radicals and decrease muscular damaged [29].

The blood increase of muscle markers, which is very common in competitive cycling races, can affect muscle function and physical performance [16,33,34,35,36]. In any case, and considering muscle biomarkers, improvements in muscle integrity in the MG were modest when compared to the CG. This observation is corroborated by the results obtained from performance tests (VO_2_max, Pmax and IEmax) where no significant differences between periods (T1 vs. T3) were observed in the same group. Thus, performance across the length of the race was not affected, allowing optimal activity in all cyclists. Based on these results, it seems that muscle status, fatigue and recovery were adequate in both groups. Therefore, it can be suggested that control of an adequate amount of Mg in the diet could be sufficient to maintain adequate muscle status and function. Supplementation can be performed when Mg-rich foods are not available or are not pleasant to certain individuals.

With respect to the hormonal response, both groups showed slight increases of serum cortisol levels at the end of the race, but levels were maintained in the physiological range. It is possible to hypothesize that an optimal body concentration of Mg could be related to adequate cortisol response, reducing stress effects in the cyclist. This seems to be consistent with other reports, in which Mg deficiency induced a systemic stress response by activation of neuro-endocrinological pathways [15,37,38].

## 5. Conclusions

In conclusion, our results indicated that Mg supplementation exceeding RDA has a modest effect in maintaining muscle integrity. In any case, adequate levels of Mg intake from diet or combined with supplements can maintain serum Mg and e-Mg levels in physiological ranges, permitting muscle recovery from intense and strenuous exercise, as is found in a cycling competition.

## Figures and Tables

**Figure 1 nutrients-11-01927-f001:**
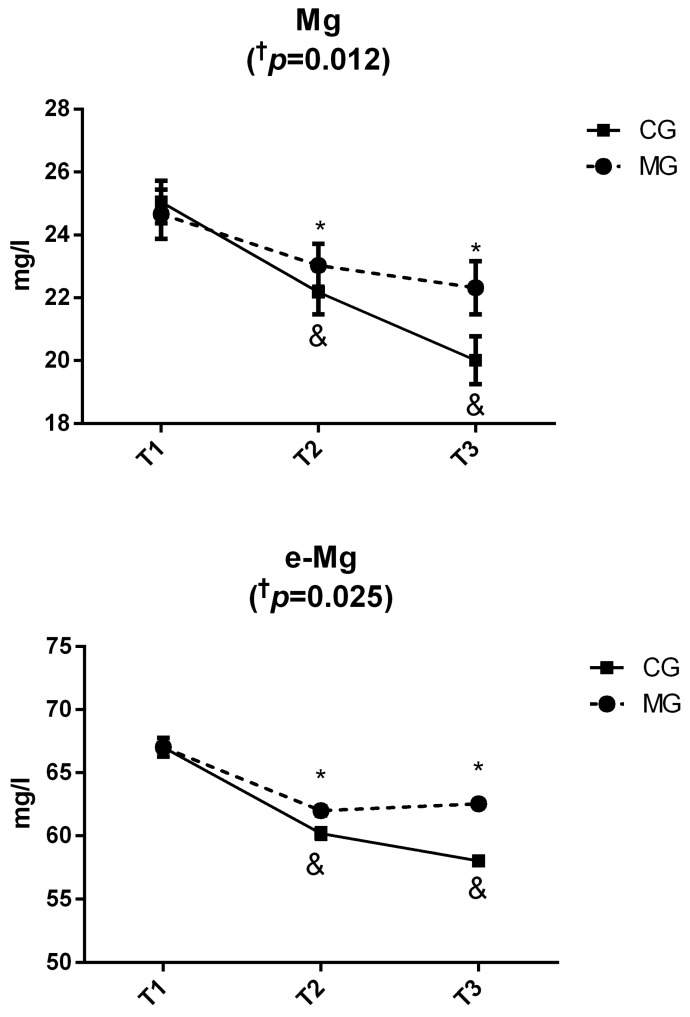
Evolution of serum magnesium (Mg) and erythrocytic Mg (e-Mg) levels throughout the study (T1, T2 and T3) in the magnesium supplemented group (MG) and the control group (CG). Data were expressed as Mean $\bar{x}$ ± SEM. ^†^ p: time × group interaction (*p* < 0.05, all such occurrences) by two-factor repeated-measures analysis of variance (ANOVA). (*) Significant differences (*p* < 0.05) between groups (CG vs. MG) in a specific time point (T1, T2 or T3) by independent *t*-test (*p* < 0.05). (&) Significant differences (*p* < 0.05) into the same group (CG or MG) at the different periods of the study compared to the beginning (T1 vs. T2 and T1 vs. T3) by Bonferroni′s test.

**Figure 2 nutrients-11-01927-f002:**
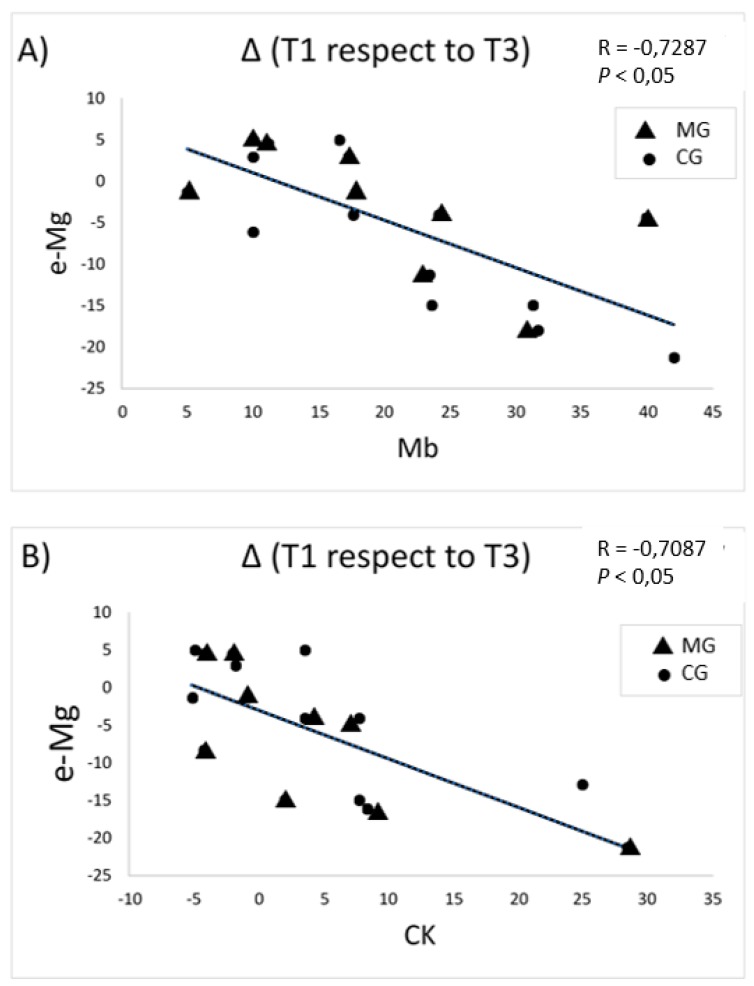
Correlation between changes of erythrocytic-Mg (e-Mg) and myoglobin (Mb) (**A**) and creatine kinase (CK) (**B**) during the stage race. Changes are expressed as differences in certain parameter at the beginning (T1) and the end (T3) of the study (Δ(T1 respect to T3)), calculated as indicated in the Materials and Methods section.

**Table 1 nutrients-11-01927-t001:** Physical and anthropometric characteristics in magnesium supplemented group (MG) and control group (CG) at the beginning of the study (T1).

	MG	CG
Age (years)	26.2 ± 1.81	26.6 ± 1.52
Weight (kg)	66.06 ± 3.59	67.1 ± 2.40
Height (cm)	176.2 ± 3.42	176.8 ± 3.052
∑6 Skinfolds (mm)	46.8 ± 0.95	46.5 ± 1.13

Data are expressed as mean ± standard error of the mean ($\bar{x}$ ± SEM).

**Table 2 nutrients-11-01927-t002:** Parameters obtained at the ergometry test in the magnesium supplemented group (MG) and control group (CG) at the beginning (T1) and at the end of the study (T3).

	T1	T3	^†^ *p*
Pmax (Watt)			
MG	571.4 ± 59.7	540.4 ± 76.9 ^§^	NS
CG	569.5 ± 54.9	539.6 ± 68.3 ^§^	
IEmax (Watt/kg)			
MG	8.66 ± 0.84	8.36 ± 1.20	NS
CG	8.62 ± 0.95	8.34 ± 0.94	
VO_2_max (mL/kg/min)			
MG	83.56 ± 2.70	83.32 ± 3.10	NS
CG	83.16 ± 2.93	83.17 ± 2.90	

Data are expressed as Mean ± Standard Error of the Mean ($\bar{x}$ ± SEM). Abbreviations used: Pmax: peak power; IEmax: Pmax/Weight; VO_2_max: maximum oxygen volume. ^†^
*p*: Results were analysed by a two-factor repeated-measures ANOVA (time × group) with no significant (NS) differences. ^§^ Significant differences (*p* < 0.05) between periods (T1 vs. T3) into the same group.

**Table 3 nutrients-11-01927-t003:** Hematological parameters of the magnesium group (MG) and control group (CG) at the beginning (T1), half race (T2) and end of the race (T3).

	T1	T2	T3	^†^ *p*
WBC (10^3^ µL^−1^)				
MG	6.02 ± 0.60	6.78 ± 0.42	9.29 ± 0.50 ^&^	>0.05
CG	6.11 ± 0.58	6.69 ± 0.62	9.59 ± 0.62 ^&^	
Platelet (10^3^ µL^−1^)				
MG	240.90 ± 18.47	203.15 ± 14.01 ^&^	249.98 ± 25.01 ^#^	>0.05
CG	245.75 ± 21.86	208.15 ± 19.01 ^&^	246.02 ± 22.13 ^#^	
RBC (10^6^ µL^−1^)				
MG	5.25 ± 0.77	4.85 ± 0.14	4.88 ± 0.28	>0.05
CG	5.21 ± 0.96	4.77 ± 0.26	4.83 ± 0.39	
Hb (g dL^−1^)				
MG	16.72 ± 0.26	14.82 ± 0.31 ^&^	15.25 ± 0.46	>0.05
CG	16.49 ± 0.46	14.90 ± 0.39 ^&^	15.08 ± 0.68	
Htc (%)				
MG	47.54 ± 0.43	46.01 ± 0.65	44.84 ± 2.63	>0.05
CG	47.86 ± 0.68	46.14 ± 0.46	45.01 ± 2.24	

Data are expressed as mean ± standard error of the mean ($\bar{x}$ ± SEM). Abbreviations used: Hb, hemoglobin; Htc, hematocrit; RBC, red blood cells; WBC, white blood cells. ^†^
*p*: time × group interaction (*p* < 0.05, all such occurrences) by two-factor repeated-measures ANOVA. (^&^) Significant differences (*p* < 0.05) in the same group (CG or MG) at different periods of the study compared to the beginning (T1 vs. T2 or T1 vs. T3) by Bonferroni’s test. (^#^) Significant differences (*p* < 0.05) in the same group (CG or MG) at different periods of the study comparing T2 vs. T3 by Bonferroni′s test.

**Table 4 nutrients-11-01927-t004:** Circulating markers of the magnesium group (MG) and control group (CG) at the beginning (T1), half race (T2) and end of the race (T3).

	T1	T2	T3	^†^ *p*
Creatinine (mg/dL)				
MG	0.86 ± 0.02	0.95 ± 0.03	1.05 ± 0.03 ^&^	>0.05
CG	0.85 ± 0.03	0.93 ± 0.04	1.03 ± 0.04 ^&^	
CK (U/I)				
MG	106.6 ± 2.3	116.2 ± 5.8	146.0 ± 5.4	>0.05
CG	104.9 ± 2.8	118.9 ± 6.2	148.2 ± 6.8	
Mb (ng/mL)				
MG	35.22 ± 1.97	35.55 ± 1.82	43.03 ± 2.45 ^&^	0.002
CG	34.89 ± 2.48	38.26 ± 2.4 *	48.64 ± 3.11 *^,&^	
LDH (U/I)				
MG	301.5 ± 41.0	357.3 ± 43.2	423.5 ± 50.2	>0.05
CG	296.6 ± 42.9	367.8 ± 48.2	418.8 ± 56.8	
GOT (U/I)				
MG	30.85 ± 3.10	46.82 ± 3.22 ^&^	46.50 ± 5.26	>0.05
CG	31.05 ± 3.55	47.04 ± 3.84 ^&^	48.48 ± 5.68	
GPT (U/I)				
MG	26.81 ± 2.85	35.12 ± 2.42 ^&^	48.98 ± 6.78 ^&^	>0.05
CG	27.01 ± 3.15	36.24 ± 3.01 ^&^	50.98 ± 6.78 ^&^	
Total Circulating Proteins (g/dL)				
MG	7.36 ± 0.10	6.95 ± 0.10 ^&^	6.99 ± 0.08 ^#^	>0.05
CG	7.25 ± 0.11	6.90 ± 0.09 ^&^	7.01 ± 0.09 ^#^	
ALD (U/I)				
MG	4.27 ± 0.58	5.36 ± 0.37	4.55 ± 0.85	>0.05
CG	4.23 ± 0.61	5.23 ± 0.43	4.44 ± 0.78	
Cortisol (µg/dL)				
MG	19.86 ± 0.91	20.54 ± 0.79	21.25 ± 0.71	>0.05
CG	19.68 ± 1.01	20.41 ± 0.85	21.16 ± 0.89	

Data are expressed as mean ± standard error of the mean ($\bar{x}$ ± SEM). Abbreviations used: ALD, aldolase; CK, creatine kinase; GOT, glutamate oxaloacetate transaminase; GPT, glutamate pyruvate transaminase; LDH, lactate dehydrogenase; Mb, myoglobin; CRE:creatinine. ^†^
*p*: time × group interaction (*p* < 0.05, all such occurrences) by two-factor repeated-measures ANOVA. (*): Significant differences (*p* < 0.05) between groups (CG vs. MG) in a specific time point (T1, T2 or T3) by independent t-test. (^&^) Significant differences (*p* < 0.05) into the same group (CG or MG) at the different periods of the study compared to the beginning (T1 vs. T2 and T1 vs. T3) by Bonferroni′s test. (^#^) Significant differences (*p* < 0.05) into the same group (CG or MG) at different periods of the study comparing T2 vs. T3 by Bonferroni’s test.

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
