# Peer review of "Impact of Magnesium Supplementation in Muscle Damage of Professional Cyclists Competing in a Stage Race"

_nutrients, 2019, doi:10.3390/nu11081927_

Round 1

Reviewer 1 Report

The study investigated the effect of Mg supplementation on muscle parameters in elite athletes. The study was randomized but not placebo controlled. Main outcome is reduced muscle damage in the Mg group and no effect of Mg supplementation on performance. it should be mentioned what Mg salt was used, Mg oxide? bioavailability of 35% (unlikely, reference?) - it is not enough to cite the producer - omit Fig.2 : use of different symbols for MG and CG would give more information. Comparable studies in elite athletes are scarce and even though the placebo is missing the results merit publication. Long term studies in athletes during a competition are not easy to perform and the beneficial results of Mg supplementation should be published.

Author Response

Dear Reviewer,

We appreciate the time you devoted to reading our manuscript and helping us to craft an improved version. We are pleased to clarify your concern which we believe will improve the impact and quality of your work. Please find below our response to your observation. We have made a concerted attempt to systematically address the specific concerns raised for this revision and we have highlighted the alterations to this revision within the manuscript in yellow for your convenience.

Reviewer 1

Point 1: The study investigated the effect of Mg supplementation on muscle on muscle parameters in elite athletes. The study was randomized but not placebo controlled.

Answer 1: The reviewer is right. We have a control group with no supplementation and a supplemented group. We have changed this point in the manuscript, identifying the “Placebo group” as “Control group”. See change in lines 96-97.

Point 2: Main outcome is reduced muscle damage in the Mg group and no effect of Mg supplementation on performance. It should mentioned what Mg salt was used, Mg oxide? Bioavailability of 35% (unlikely, reference?). It is not enough to cite the producer.

Answer 2: Thank you for your comment. The product used contains pure Mg according to the manufacturer (line 105). We estimated a bioavailability of 35%, taking into account the diet composition of participants and following the data provided by the experts. We included a new reference regarding this point: reference 17. The number of references has been changed accordingly.

Point 3: Fig 2: use of different symbols for MG and CG would give more information.

Answer 3: Thank you for your recommendation. The authors have used triangles to indicate the Mg group and even circles of the control group.

Point 4: Comparable studies in elite athletes are scarce and even though the placebo is missing, the results merit publication. Long term in athletes during a competition are not easy to perform and the beneficial results of Mg supplementation should be published.

Answer 4: We agree with the reviewer.

The English has been revised by Dr Jonathan Jones, a native American scientist that uses to speak fluent scientific English.

Reviewer 2 Report

This study seeks to determine the potential for Mg supplementation in mitigating indicators of muscle damage in competitive cyclists competing in a 3 week long stage race. Within the limitations of the study, it does demonstrate the physiological effectiveness of Mg supplementation and modest effects of such supplementation on one indicator of muscle damage. The most noteworthy finding is the modest correlation between e-Mg levels and two indicators of muscle damage in serum (CK & Mb). The conclusions are appropriate.

One issue the authors need to address is the reasons why the control group did not receive a placebo. Since the use of a placebo is typical in these types of studies and serves an important purpose in optimal experimental design, the authors should explain why the control group was not a "placebo" group and how this might have influenced the study results.

Author Response

Dear Reviewer,

We appreciate the time you devoted to reading our manuscript and helping us to craft an improved version. We are pleased to clarify your concern which we believe will improve the impact and quality of your work. Please find below our response to your observation. We have made a concerted attempt to systematically address the specific concerns raised for this revision and we have highlighted the alterations to this revision within the manuscript in yellow for your convenience.

Reviewer 2

Point 1: This study seeks to determine the potential for Mg supplementation in mitigating indicators of muscle damage in competitive cyclists competing in a 3 week long stage race. Within the limitations of the study, it does demonstrate the physiological effectiveness of Mg supplementation and modest effects of such supplementation on one indicator of muscle damage. The most networthy finding is the modest correlation between e-Mg levels and two indicators of muscle damage in serum (CK & Mb). The conclusions are appropriate. One issue the authors need to address is the reasons why the control group did not receive a placebo. Since the use of a placebo is typical in these types of studies and serves an important purpose in optimal experimental design, the authors should explain why the control group was not a “placebo” group and how this might have influenced the study results.

Answer 1: The reviewer is completely right. Nevertheless, the main problem we had to face was that the participants were competing in a long stage cycling race that is a reference in the world after the “Tour de France” and “Giro d’Italia”: “Vuelta ciclista a España”. Therefore, one point to respect with the participants was to avoid disturbances regarding the progression of the competition, because a big amount of money was invested by both teams. In this respect, one team accepted to follow the supplementation protocol, but in the other case, coaches were reluctant to accept a placebo. Altogether, we decided to give nothing and follow the evolution of cyclists. We think that this approach is a good option, taking into account that a “placebo” was not possible. In addition, since both teams were independent, no change of information occurred between participants and we guest that the influence in the study was nil.

The English has been revised by Dr Jonathan Jones, a native American scientist that uses to speak fluent scientific English.
